# Motor Development Comparison between Preterm and Full-Term Infants Using Alberta Infant Motor Scale

**DOI:** 10.3390/ijerph20053819

**Published:** 2023-02-21

**Authors:** Jooyeon Ko, Hyun Kyoon Lim

**Affiliations:** 1Department of Physical Therapy, Daegu Health College, Daegu 41453, Republic of Korea; 2Korean Balance Ability Data Center, Daegu 41453, Republic of Korea; 3Medical Measurement Team, Korea Research Institutes of Standards and Science, Daejeon 34113, Republic of Korea; 4Department of Medical Physics, University of Science and Technology, Daejeon 34113, Republic of Korea

**Keywords:** Alberta Infant Motor Scale (AIMS), motor development, preterm infants, midline development, motor skills

## Abstract

The Alberta Infant Motor Scale (AIMS) was developed to evaluate the motor development of infants up to 18 months of age. We studied 252 infants in three groups (105 healthy preterm infants (HPI), 50 preterm infants with brain injury (PIBI), and 97 healthy full-term infants (HFI) under 18 months, corrected age (CoA)) using AIMS. No significant differences were found among HPI, PIBI, and HFI in infants less than 3 months old, yet significant differences were noted in positional scores (*p* < 0.05) and total scores for those four to six months of age and seven to nine months of age. A significant difference was also found in standing items for infants over 10 months (*p* < 0.05). After four months, there was a difference in motor development between preterm (with and without brain injury) and full-term infants. In particular, there was a significant difference in motor development between HPI and HFI and between PIBI and HFI at four to nine months, when motor skills developed explosively (*p* < 0.05). After four months, motor developmental delays (10th ≥) were observed in HPI and PIBI at rates of 26% and 45.8%, respectively. Midline supine development, a representative indicator of early motor development, was slower even in healthy preterm infants than in full-term infants. AIMS has a good resolution to discriminate preterm infants who are showing insufficient motor development from 4 months to 9 months.

## 1. Introduction

Annually, worldwide, approximately two to five in every 1000 babies are born premature [1]. Significant and slightly lower motor skill development is possible in a preterm baby, even in cases of no or minor brain damage [2,3]. Motor development in infancy occurs rapidly, and early intervention is required in cases of immature or abnormal motor skills [4,5]. Earlier intervention often produces better results [6]. Direct, quantitative, objective, and infant-friendly observation methods are needed for early detection [7].

Motor assessment tools have been used for discrimination, prediction, and/or evaluation [8]. Discrimination compares the current level of motor development with a reference group [9]. Future performance could be predicted by classifying infants based on potential future status [10]. An evaluative tool is used to measure the degree of change in motor function over time or after an intervention. Few measuring tools have all three of these features [11].

The Alberta Infant Motor Scale (AIMS) was developed to discriminate and evaluate the motor development of infants aged 0 to 18 months [12]. The AIMS has been widely used for research and clinical purposes. The AIMS positional scores and total scores have been used to compare motor development between preterm and full-term infants, including their reliability [13]. As infants motor development features could vary, how well they did each item should be checked for an accurate AIMS measurement and evaluation [14]. Weight-bearing, posture, and antigravity movement are key descriptors of motor skills used in the AIMS. The motor components are described in the score sheet and manual [15]. How preterm and full-term infants accomplish each AIMS item and how the score results are utilized have not been well studied [12,16].

The objectives of this study are to analyze four positions and total scores, to investigate motor delay, and to describe the representative motor skills of healthy preterm infants (HPI), preterm infants with brain injury (PIBI), and healthy full-term infants (HFI) using the AIMS.

## 2. Materials and Methods

### 2.1. Participants

We recruited 252 infants using a social networking service, including preterm infants (105 healthy preterm infants (HPI), 50 preterm infants with brain injury (PIBI)), and 97 healthy full-term infants (HFI) under 18 months, either chronological or corrected age, from seven cities in South Korea from 2017 to 2019. All parents signed consent forms before participation (IRB no. 2-1040781-AB-N-01-2017102HR). The selection criteria of the preterm infants were gestational age < 37 weeks, birth weight < 2500 g, no genetic syndrome or congenital brain malformation, and no or mild brain events (intraventricular hemorrhage grade I or IIa or periventricular densities lasting less than seven days) at serial scan. For full-term infants, selection criteria were gestational age between 38–42 weeks, birth weight ≥ 2500 g, and no neurological or musculoskeletal conditions. The general characteristics of the subjects are summarized in Table 1.

### 2.2. Evaluation Tool (AIMS)

The AIMS evaluates infant motor development quantitatively and qualitatively. Specifically, it focuses on the variety and multitude of specific motor skills contributing to an infant’s motor repertoire or the fundamental components contributing to a specific motor skill. The total assessment time was about 20–30 min. The AIMS consists of 58 items in four basic functional positions: 21 prone items, 9 supine items, 12 sitting items, and 16 standing items. The most mature and least mature observed items in each of the four positions represent the infant’s possible motor repertoire in that position or his, or her “window” of current skills. Each item is scored as either observed (score 1) or not observed (score 0). The items observed were added to determine an infant’s total AIMS score. The AIMS raw scores, which ranged from 0 to 58, were converted to a percentile ranking [7]. We used the AIMS scoresheet and manual that was translated into Korean for data collection. We used the Canadian AIMS normative sample value as there is no Korean norm value to identify the percentile ranking of the infants’ overall motor performance.

### 2.3. Procedures

This was a cross-sectional, quantitative, and comparative study. The 252 infants were classified into three independent groups (HPI, PIBI, and HFI) and were assessed to collect data for four independent age groups: zero to three months, four to six months, seven to nine months, and 10 months ≤. The AIMS was administrated in all three groups. Parents were interviewed to obtain the infants’ general and medical histories. The AIMS is an observational assessment tool that requires minimal infant handling by the examiner. Although some items are required to do a specific prompt, facilitation and handling were consciously avoided when assessing an infant. For the AIMS assessment, one well-trained physical therapist who possessed more than ten years of infant motor development and AIMS testing experience performed the AIMS examination and score evaluation to maintain measurement consistency. The AIMS tests were performed in a warm, quiet room with the parent or caregiver. During the assessment, the infants were in an active state, and they wore simple clothes for the truncal posture evaluation. The infants were encouraged to demonstrate the motor skills they were able to spontaneously perform without the assistance of the examiner or parent. All assessments were recorded with a smartphone camera. And then, video recordings were analyzed to obtain positional and total scores for each infant.

### 2.4. Data Analysis

We used IBM SPSS statistics 27.0.1.0 for Windows for the statistical analysis. We calculated descriptive statistics of the subjects’ general characteristics. As the AIMS data distribution was not normal, we used non-parametric tests for data comparison. We checked the differences between gestational age (<32 weeks vs. ≥32 weeks) and birth weight (<1500 g vs. ≥1500 g). We used the Kruskal–Wallis test and the Mann–Whitney U-test to compare the significant differences between the groups for prone, supine, sitting, and standing positions and total scores. To determine the motor developmental delay, the 10th percentile was established as the cutoff point [17,18,19]. To classify motor performance, we adopted five percentile ranges: 0–10th percentile for a motor developmental delay, 11th–25th percentile for suspect performance, 26–75th percentile for normal performance, 76–90th percentile for very good performance, and 91st–100th percentile for excellent performance [20]. The motor repertories were investigated by identifying the motor skills performed by more than 90% of infants aged 0–12 months, and the most mature observed window item was analyzed as the representative motor skill. The significance level was set at *p* < 0.05.

## 3. Results

### 3.1. Alberta Infant Motor Scale (AIMS) Positional and Total Scores

We found no significant differences among the HPI, PIBI, and HFI in zero to three-month-old infants. We found significant differences for the four positional scores (prone, supine, sitting, and standing) and total scores for four- to six-month and seven- to nine-month-old infants among the HPI, PIBI, and HFI groups (*p* < 0.05). Preterm infants showed significantly lower scores in motor development compared to healthy full-term infants (Mann–Whitney U-test, *p* < 0.05). Especially, all four positional scores and total scores of seven- to nine-month-old infants showed a significant difference between healthy preterm infants and preterm infants with mild brain events. Significant differences were also found between HPI vs. HFI, and PIBI vs. HFI for the standing scores for infants older than ten-month-old infants (Table 2). No significant difference was found in gestational age (<32 weeks vs. ≥32 weeks) and birth weight (<1500 g vs. ≥1500 g).

### 3.2. Identification of Motor Delay and Classification of Motor Performance

Both healthy preterm infants and preterm infants with brain injuries showed significant motor developmental delays. In particular, PIBI presents early in life. No motor development delay was observed in healthy full-term infants (Table 3). Figure 1 shows the distribution of the motor developmental delay (0–10th), suspect (11th–25th), normal (26–75th), very good (76–90th), and excellent performance (91st–100th) among the independent three groups aged zero to three months, four to six months, seven to nine months, and 10 months ≤.

### 3.3. Motor Repertories and Representative Motor Skills

Table 4 shows motor repertories and representative motor skills from 1–12 months for HPI, PIBI, and HFI. Motor skill items are added with increasing age. It was easy to understand how motor skills developed for prone, supine, sitting, and standing for each month using the motor skill acquisition.

## 4. Discussion

This study analyzed motor features such as motor performance, motor delay, categorization of motor performance, motor repertories, and representative motor skills of infants born preterm, both with and without brain injury, and full-term infants. Based on the AIMS data, differences were demonstrated in overall motor performance among healthy preterm infants, preterm infants with mild brain injury, and healthy full-term infants at ages four to six months and seven to nine months, in all positional and total scores.

Muscle strength imbalances between extensors and flexors [13], gross motor developmental differences [21], and late walking initiation age [22] are features of preterm infants. In our results, there were significant differences in the score of standing position from ten months onward, even between low-risk preterm infants and full-term infants. Preterm infants generally show delayed motor development in the onset of upright locomotion [23].

The AIMS has been used to find infants with motor developmental delays. The cutoff points for motor development delay used the 5th percentile [24] or the 10th percentile [25,26] as a threshold. In this study, the 10th percentile was used as suggested by Darrah et al. [17]. We may obtain higher specificity values if we use the 5th percentile, although we may not detect as many infants with motor delays. We wanted to detect as many infants with even minor motor development delays as possible [17].

One previous study showed 19 (22%) out of 87 infants (preterm infants born at ≤30 weeks gestational age) as having a motor development delay from an AIMS evaluation at four months old using the 10th percentile as the threshold [11]. Another study showed 37 (37%) out of 100 infants (preterm infants born at ≤32 weeks gestational age) as having a motor development delay from an AIMS evaluation at four months, when using the 10th percentile as the threshold [27]. In this study, we found 10 (17%) out of 58 infants (full-term and preterm infants born with mild brain events) at four to six months of age and two (6%) out of 34 infants aged zero to three months. Our study covers only low-risk preterm infants, while the previous studies [11,24] covered high-risk preterm infants, including IVH grades III and IV. The gestation period was different in the three studies. We found no significant score differences between ≤32 weeks and >32 weeks. No motor development delay was found in full-term infants (0–18 months, n = 97), whereas motor development delays were found in all preterm infants (HPI and PIBI).

For the motor performance distribution by the AIMS five percentile range, a normal general distribution motor development score was found in HPI and HFI, while motor delay and suspect were frequently observed in PIBI. It was hard to find a categorization study on motor performance using AIMS. One study classified preterm and full-term infants (total n = 24) into five categories [20] and reported the highest AIMS frequency of normal distribution at the six-month evaluation when the infants were evaluated at four and six months. Our study also showed similar results in that all full-term infants participating in this study showed normal results, except one four-month-old infant, while many of the preterm infants showed motor delays and suspicion of motor delays.

To date, the AIMS study mainly used positional and total scores to determine the motor development status of infants. It is, however, more important to measure each item precisely, to establish a profile, and to apply the results to infants who need early intervention systemically and efficiently [28,29]. In this study, we compared three groups (HPI, HFI, and PIBI) using motor skill repertories that over 90% of infants (0–12 months old) fully learned in four positions. In prone development, we observed a significant difference at nine months between preterm and full-term infants in motor repertories. The HFI group showed the most variety in antigravity movements. In supine development, the HFI group learned nine motor repertory items within six months, while HPI learned them in eight months, and PIBI in nine months. Preterm infants showed significantly different motor skills in the sitting position from five to nine months compared to full-term infants. These results are similar to those reported in previous studies [13,17,30,31,32].

Motor trajectories were compared between preterm (gestational period ≤ 29 weeks) and full-term infants using the positional score and total AIMS score [13], demonstrating that the sitting position of four to eight-month-old infants differed the most between preterm and full-term infants. We also found different motor development skills for the standing position between the two groups for their motor repertories. We could not confirm a difference in mobility development, such as cruising skills. The AIMS evaluates motor skills qualitatively, and motor skill quality may affect motor development quantitatively in infant motor learning. Even though low-risk preterm infants showed good quality of motor repertories, similar to full-term infants beyond 10 months, except for standing development, a periodic evaluation using a proper measurement tool such as the AIMS is highly recommended for preterm infants. The AIMS could be used as a guideline for play-based home motor skill learning exercises [33] and early development interventions [34].

Regarding the representative motor skills by month, we identified the most mature observed item in the window, as indicated in the AIMS manual [7]. We also identified the representative motor skill monthly and apply this to evaluate motor skill perfection. The AIMS and several infant motor assessment tools, such as the IMP [28], the GMA [35], and the SINDA [32] include measurement items to evaluate midline development in supine in the first months of life. Early motor development is characterized by a progressive organization of arm and leg movements toward the midline [36]. Lack of midline indicates that the child lacks the ability to bring their limbs toward the center of their body, so the limbs remain close to the supporting surface, their overall appearance looks flat, and their movements are inactive [37,38]. In the AIMS, supine item four indicates midline development (weight symmetrically distributed on the head, trunk, and buttocks, head in the midline with the chin tuck, and bringing hands to the midline). In this study, only the HFI group achieved the item at five months, which means supine item four is the representative motor skill of that age group. A total of 90% of Canadian norm samples performed this item at approximately four months [7]. Korean infants tend to spend more time in supine than prone or side lying, which is a cultural and parenting habit and could affect slower midline development than in infants in Western countries [39,40]. In our study, in the HPI and PIBI groups, the representative motor skill for five months was supine item three, in which the infant may bring two hands close to their trunk, yet be unable to get their two hands at their midline. The World Health Organization (WHO) selected six gross motor development milestones, and independent sitting without support was considered the first major skill [41]. Items 8 and 9 were used to measure playing ability while sitting without support was the AIMS sitting position. HFI learned items 8 and 9 by the ninth month, whereas HPI learned them after the ninth month (90%). Our study supports the previous study results by Pin [13], where the preterm infant showed slower motor development for the stable sitting posture because of a muscle force imbalance between the flexion and extension muscles. Considering the variability in the development of motor skill sequences and emergence timing, the data of the individual AIMS item over time (1–18 months) would be very meaningful in both research and clinical practices.

## 5. Conclusions

In this study, we found significant motor development differences for four positions (prone, supine, sitting, and standing) among healthy preterm infants (HPI), healthy full-term infants (HFI), and preterm infants with mild brain injury (PIBI), from four to nine months (CoA). We were able to understand the motor development status precisely, with the AIMS item-based approaches. We propose representative motor repertories and the most matured motor skills month-by-month using an AIMS based on each item score and total scores for detailed observation, critical evaluation, and suitable intervention.

## 6. Limitations of the Study

We could not investigate changes in motor function over time through follow-up assessments. We also had only one infant with PIBI after 10–12 months. As the infant had a very mild brain injury, the infant showed similar motor behavior compared to HFI. Finally, there were not enough infants used for the 10–12-month period in this study.

## Figures and Tables

**Figure 1 ijerph-20-03819-f001:**
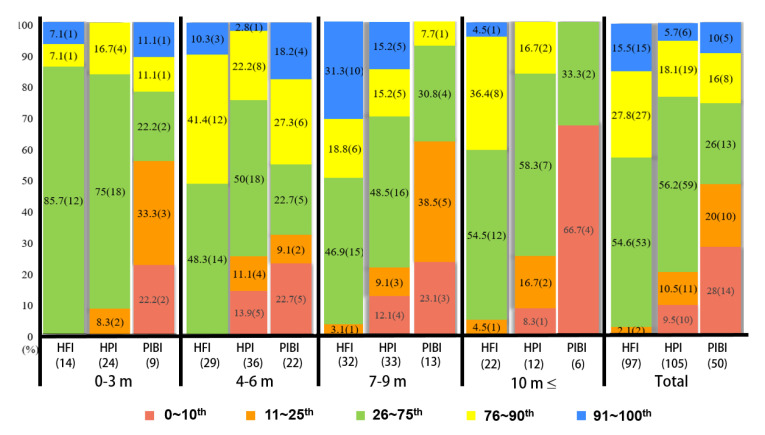
Distribution of five percentile categories of motor performance using the AIMS total score among three groups. HPI: healthy preterm infants, PIBI: preterm infants with brain injury, and HFI: healthy full-term infant. Age was calculated with corrected age or chronological age.

**Table 1 ijerph-20-03819-t001:** The general characteristics of the subjects.

	HFI Group	HPI Group	PIBI Group
CoA or CA (days)	201.9 ± 98.9	169.5 ± 100.1	179.6 ± 120.1
Gestational age (days)	278.0 ± 6.0	223.6 ± 20.6	203.3 ± 26.3
Gestational weeks			
Less than 32 weeks More than 32weeks	- -	48 (45.7) 57 (54.3)	37 (74.0) 13 (26.0)
Birth weight (grams)	3321.2 ± 145.3	1617.8 ± 704.9	1295.0 ± 638.5
Birth weight (grams)			
Less than 1500 g More than 1500 g		47 (44.8) 58 (55.2)	31 (62.0%) 19 (38.0%)
Gender			
Girls Boys	35 (36.1) 62 (63.9)	51 (48.6) 54 (51.4)	21 (42.0) 29 (58.0)
Brain injury			
PVL			
Yes No	0 (0.0) 97 (100.0)	0 (0.0) 105 (100.0)	24 (48.0) 26 (52.0)
IVH (grade I or IIa)			
Yes No	0 (0.0) 97 (100.0)	0 (0.0) 105 (100.0)	35 (70.0) 15 (30.0)
Age bands (months)			
0–3 4–6 7–9 10≤ Total	14 (14.4) 29 (29.9) 32 (33.0) 22 (22.7) 97 (100.0)	24 (22.9) 36 (34.3) 33 (31.4) 12 (11.4) 105 (100.0)	9 (18.0) 22 (44.0) 13 (26.0) 6 (12.0) 50 (100.0)

HPI: healthy preterm infants, PIBI: preterm infants with brain injury, HFI: healthy full-term infant, Unit: mean ± standard deviation or n (%), CoA: corrected age, CA: chronological age, PVL: periventricular leukomalacia, IVH: intraventricular hemorrhage.

**Table 2 ijerph-20-03819-t002:** Alberta Infant Motor Scale (AIMS) positional and total scores differences by groups at four different ages (n = 252).

	0–3 m			4–6 m			7–9 m			10 m ≤		
	N	Mean (Min–Max)	*p*	n	Mean (Min–Max)	*p*	n	Mean (Min–Max)	*p*	n	Mean (Min–Max)	*p*
Prone												
HFI	14	2.5 (2–5)		29	7 (4–21)		32	15 (6–21)		22	21 (21–21)	
HPI	24	3 (1–5)	0.988	36	6 (2–18) ^a,b^	0.000 *	33	12 (6–21) ^a^	0.005 *	12	21 (19–21)	0.205
PIBI	9	2 (1–7)		22	5 (1–10) ^c^		13	10 (2–14) ^c^		6	21 (15–21)	
Supine												
HFI	14	3 (3–5)		29	9 (3–9)		32	9 (9–9)		22	9 (9–9)	
HPI	24	3 (2–4)	0.149	36	5 (2–9) ^b^	0.000 *	33	9 (4–9) ^b^	0.034	12	9 (9–9)	1.000
PIBI	9	3 (2–9)		22	4 (2–9) ^c^		13	9 (4–9) ^c^		6	9 (9–9)	
Sitting												
HFI	14	1 (1–2)		29	4 (2–12)		32	10 (5–12)		22	12 (12–12)	
HPI	24	1 (1–2)	0.645	36	3 (1–10) ^b^	0.000 *	33	9 (3–12) ^a,b^	0.000 *	12	12 (12–12)	0.059
PIBI	9	1 (1–3)		22	2 (1–6) ^c^		13	5 (0–12) ^c^		6	12 (6–12)	
Standing												
HFI	14	2 (2–3)		29	3 (2–10)		32	3 (3–10)		22	11 (8–16)	
HPI	24	2 (1–2)	0.062	36	2 (2–6) ^b^	0.000 *	33	3 (2–10) ^a,b^	0.000 *	12	10 (4–16) ^b^	0.135
PIBI	9	2 (1–2)		22	2 (2–3) ^c^		13	3 (1–3) ^c^		6	10.5 (2–14) ^c^	
Total score												
HFI	14	9 (8–13)		29	22 (11–52)		32	39 (29–52)		22	53 (50–58)	
HPI	24	9 (6–25)	0.594	36	17 (7–43) ^a,b^	0.000 *	33	33 (16–52) ^a,b^	0.000 *	12	51.5 (46–58) ^b^	0.093
PIBI	9	8 (6–21)		22	14 (6–26) ^c^		13	27 (7–36) ^c^		6	52.5 (32–56)	

* *p* < 0.05, mean (range: min–max), HPI: healthy preterm infants, PIBI: preterm infants with brain injury, HFI: healthy full-term infant, data were compared using Kruskal–Wallis test, followed by post hoc comparisons by Mann–Whitney U-test. ^a^ HPI vs. PIBI. ^b^ HPI vs. HFI. ^c^ PIBI vs. HFI.

**Table 3 ijerph-20-03819-t003:** Motor developmental delay using AIMS ≤ 10th by groups and ages.

AIMS ≤ 10th			
Age	HFI	HPI	PIBI
0–3 m	0/14 (0.0)	0/25 (0.0)	2/9 (22.2)
4–6 m	0/29 (0.0)	5/36 (13.9)	5/22 (22.7)
7–9 m	0/32 (0.0)	4/33 (12.1)	3/13 (23.1)
10 m ≤	0/22 (0.0)	1/12 (8.3)	4/6 (66.7)
Total	0/97 (0.0)	10/105 (9.5)	14/50 (28.0)

**Table 4 ijerph-20-03819-t004:** Motor repertories and representative motor skills through 1–12 months using AIMS (n = 234).

Age	Group	n	Prone (Items 1–21)	Supine (Items 1–9)	Sitting (Items 1–12)	Standing (Items 1–16)
1 m	HFI	0				
	HPI	4	**1. prone lying (1)**	1. supine lying (1), **2. supine lying (2)**	**1. sitting with support**	**1. supported standing (1)**
	PIBI	0				
2 m	HFI	8	1, **2**	1, 2, **3. supine lying (3)**	**1**	1, **2**
	HPI	6	1, **2. prone lying (2)**	1, **2**	**1**	1, **2. supported standing (2)**
	PIBI	2	1, **2**	1, **2**	**1**	1, **2**
3 m	HFI	6	1, **2**	1, 2, **3**	**1**	1, **2**
	HPI	14	1, **2**	1, 2, **3**	**1**	1, **2**
	PIBI	7	**1**	1, **2**	**1**	**1**
4 m	HFI	4	1, 2, 3, **4**	1, 2, **3**	**1**	1, **2**
	HPI	14	1, **2**	1, 2, **3**	**1**	1, **2**
	PIBI	10	1, 2, 3. prone prop, **4. forearm support (1)**	1, 2, **3**	**1**	1, **2**
5 m	HFI	15	1, 2, 2, 4, 5, **6. forearm support (2)**	1, 2, 3, **4. supine lying (4)**	1, 2. sitting with propped arms, **3. pull to sit**	1, **2**
	HPI	14	1, 2, 3, 4, **5. prone mobility**	1, 2, **3**	**1**	1, **2**
	PIBI	9	1, 2, **3**	1, 2, **3**	**1**	1, **2**
6 m	HFI	10	1, 2, 3, 4, 5, **6**	1, 2, 3, 4, 5. hands to knees 6. active extension, 7. hands to feet, 8. rolling supine to prone without rotation, **9. rolling supine to prone with rotation**	1, 2, 3, 4, **5**	1, 2, **3. supported standing (3)**
	HPI	8	1, 2, 3, 4, 5, **6**	1, 2, **3**	1, 2, **3**	1, **2**
	PIBI	3	1, 2, 3, 4, 5, **6**	1, 2, 3, **4**	1, 2, 3, 4.unsustained sitting, **5. sitting with arm support**	1, **2**
7 m	HFI	10	1, 2, 3, 4, 5, **6**	1, 2, 3, 4, 5, 6, 7, 8, **9**	1, 2, 3, 4, 5, **6. unsustained sitting without arm support**	1, 2, **3**
	HPI	16	1, 2, 3, 4, 5, **6**	1, 2, 3, **4**	1, 2, **3**	1, **2**
	PIBI	3	1, 2, 3, 4, 5, **6**	1, 2, 3, 4, **8**	1, 2, **3**	1, **2**
8 m	HFI	18	1, 2, 3, 4, 5, 6, 7. extended arm support, **9. swimming**	1, 2, 3, 4, 5, 6, 7, 8, **9**	1, 2, 3, 4, 5, 6, 7, **8. sitting without arm support (1)**	1, 2, **3**
	HPI	7	1, 2, 3, 4, 5, 6, 10. reaching with forearm support, **11. pivoting**	1, 2, 3, 4, 5, 6, 7, 8, **9**	1, 2, 3, 4, 5, 6, **7. weight shift in unsustained sitting**	1, 2, **3**
	PIBI	9	1, **2**	1, 2, 3, **4**	-	**1**
9 m	HFI	4	1, 2, 3, 4, 5, 6, 7, 8. rolling prone to supine without rotation, 9, 10, 11, 12. rolling prone to supine with rotation, **13. 4-point kneeling (1)**	1, 2, 3, 4, 5, 6, 7, 8, **9**	1, 2, 3, 4, 5, 6, 7, 8, 9. reach with rotation in sitting, **11. sitting to 4-point kneeling**	1, 2, **3**
	HPI	10	1, 2, 3, 4, 5, 6, 7, **8**	1, 2, 3, 4, 5, 6, 7, 8, **9**	1, 2, 3, 4, 5, 6, **7**	1, 2, **3**
	PIBI	1	1, 2, 3, 4, 5, 6, **7**	1, 2, 3, 4, 5, 6, 7, 8, **9**	1, 2, 3, 4, **5**	1, 2, **3**
10–12 m	HFI	16	1, 2, 3, 4, 5, 6, 7, 8, 9, 10, 11, 12, 13, 14, 15, 16, 17, 18, 19, 20, **21**	1, 2, 3, 4, 5, 6, 7, 8, **9**	1, 2, 3, 4, 5, 6, 7, 8, 9, 10, 11, **12**	1, 2, 3, 4, 5, 6, 7, 8, **10**
	HPI	5	1, 2, 3, 4, 5, 6, 7, 8, 9, 10, 11, 12, 13, 14. propped side lying, 15. reciprocal crawling, 16.4-point kneeling to sitting or half-sitting, 17. reciprocal creeping (1), 18. reaching from extended arm support, **19. Four-point kneeling (2)**	1, 2, 3, 4, 5, 6, 7, 8, **9**	1, 2, 3, 4, 5, 6, 7, 8, 9, 10.sitting rone, 11, **12. sitting without arm support (2)**	1, 2, 3, **4. pulls to stand with support**
	PIBI	1	1, 2, 3, 4, 5, 6, 7, 8, 9, 10, 11, 12, 13, 14, 15, 16, 17, 18, 19, 20. modified 4-point kneeling, **21. reciprocal creeping (2)**	1, 2, 3, 4, 5, 6, 7, 8, **9**	1, 2, 3, 4, 5, 6, 7, 8, 9, 10, 11, **12**	1, 2, 3, 4, 5.pull to stand/stands, 6.supported standing with rotation, 7. cruising without rotation, 8. half-kneeling, 9. controlled lowering through standing, **10. cruising with rotation**

HPI: healthy preterm infants, PIBI: preterm infants with brain injury, and HFI: healthy full-term infant. In this table, a number means an item that was accomplished by over 90% of all infants. The number in bold type indicates the most mature observed item of the window (representative motor skill).

## Data Availability

Data sharing is not available.

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
