# Peer review of "Motor Development Comparison between Preterm and Full-Term Infants Using Alberta Infant Motor Scale"

_ijerph, 2023, doi:10.3390/ijerph20053819_

Round 1

Reviewer 1 Report

Thank you for the opportunity to review this manuscript. I think this paper needs a revision; I added my comments in the attached file. 

Author Response

Dear reviewer,

We appreciate the reviewer's good comments on the MS. Total of eight comments were giving to this MS. Here are our responses. All line numbers are based on newly revised MS. Correction

#1 .

A. I presume it was the corrected age for the preterm group, but this is not said.
<Response> We corrected it (please see line 14)

B. But the full term babies were, I presume, a control group, or gold standard, for evaluating the use of AIMS Scale for preterm motor development. I am not sure that this conclusion is pertinent after reading the entire manuscript.
<Response> We corrected it (please see line 24)

#2 Introduction: No comments for correction

<Response> N/A

#3 Material and methods: No comments for correction

<Response> N/A

#4 Analysis: I think that this resulted in the data in Table 4. I am afraid that I could not understand the objective of this analysis. Were you looking for predictive skills of normality? Or simply searching the most frequent skills? But, could you really say that these skills?are?“representative”? Would the lack of appearence of these skills means risk of delay? To affirm such a thing you had to associate each item to a normal pattern in the future – that is, another analysis, of sensitivity and sensibility.

<Response> As indicated in the caption of the Table 4, we wanted to confirm the most frequent motor skills of infants. We did not evaluate AIMS for its sensitivity and specificity in this study. Instead, we like to mention the cultural difference between eastern and western countries. We added the following sentences in the main text with two references (see line 244~246).

Korean Infants tend to spend more time in supine than prone or side lying, which is cultural and parenting habit and could affect the slower midline development than infants in west countries [39, 40].

#5 Results: No comments for correction

<Response> N/A

#6 Table 2: The authors display the results of the scores in AIMS evaluation. They state in the footnotes of the table that they are presenting Mean (SD); but in the table it is written: M (min-max). Is it mean (min-max), median (min-max)? I could not identify the SD.

<Response> Corrected. (please see Table 2, line 137~138)

#7 Figure 1: This figure, although showing interesting findings, is not clear.

<Response> We added caption for the color. Now Fig.1 has been improved.

#8 Table 4: This is a very confusing table, I think it deserves summarization. Which is the meaning of these achievements?

<Response> We appreciate the concern and attention of the reviewer. However, we believe the Table 4 is well organized and understandable to readers. Specifically, we tried to show the development of the 58 items of the AIMS from one to 12 months of all subjects in this study. Most of papers regarding the AIMS studies are focusing on subtotal scores or total scores majorly not individual items.

#9 Discussion: It is based in the descriptions of table 4, but I was

uncertain about the analysis: is the frequency of the presence of a skill robust enough to support these conclusions?

<Response> Authors appreciate the detail comments on discussion of this manuscript by the reviewer. Conclusion was not only based on Table 4. Authors prepared Table 2 ~ Table 4 to reach the conclusion and discussion thoroughly. We think AIMS is robust enough to draw our conclusion.

Table 2 is showing universal agreement on motor development between preterm and fullterm infants. Table 3 is showing motor development delay based on AIMS manual. In their knowledge, authors believe Table 4 is a new trial which is showing the motor repertories motor and representative motor skills by the individual items of AIMS as well as by subtotal or total scores. The following sentence is added in the main text (Line 257~259)

Considering variability in the development of motor skill sequences and emergence timing, the data of the individual AIMS item over time (1 ~ 18 months) would be very meaningful in both research and clinical practice.

In addition, we checked all English language.

Reviewer 2 Report

The authors analyze AIMS four positions domains and total scores, to investigate motor delay, and to describe representative motor skills of the HPI, PIBI, and HFI. The article is easy to follow. However, the paper would be furtherstrengthened by adressing some major or minor issues:

1/ The conclusion was significant motor development difference between three groups. This result was already expected. We are waiting to another interpretation of these results. 

2/ It is important to know the gestational age at birth and the number of extremely preterm infants.

3/ Also, it is important to mentionned the incidence of the infants small for gestational age.

Author Response

The authors analyze AIMS four positions domains and total scores, to investigate motor delay, and to describe representative motor skills of the HPI, PIBI, and HFI. The article is easy to follow. However, the paper would be furtherstrengthened by adressing some major or minor issues:

1/ The conclusion was significant motor development difference between three groups. This result was already expected. We are waiting to another interpretation of these results. 

<Response> As indicated in the caption of the Table 4, we wanted to confirm the most frequent motor skills of infants. We did not evaluate AIMS for its sensitivity and specificity in this study. Instead, we like to mention the cultural difference between eastern and western countries. We added the following sentences in the main text with two references (see line 244~246).

Korean Infants tend to spend more time in supine than prone or side lying, which is cultural and parenting habit and could affect the slower midline development than infants in west countries [39, 40].

2/ It is important to know the gestational age at birth and the number of extremely preterm infants. 

<Response> Gestational age information is already in the main text (please see line 63 to 66, and Table 1).

3/ Also, it is important to mentionned the incidence of the infants small for gestational age.

<Response> We could not get the point of comment by the reviewer. If the reviewer want to know weight information, please make a reference with Table 1 also. (please see Table 1).

English of our manuscript was all corrected by a native speaker again.

Round 2

Reviewer 1 Report

Environment Research and Public Health

 Manuscript ID ijerph-2153274

Title: Motor Development Comparison between Preterm and Fullterm Infants using Alberta Infant Motor Scale

Second revision

 Thank you for the opportunity of revising the authors’ answers, which I read carefully. The manuscript improved a lot, and I am satisfied with the explanations. Neverheless, I still have two points to consider, which I am stating below.

In relation to table 2, thank you for the correction, and Figure 1 is now very clear.

I read Table 4 with the observations of the authors in mind: it is still a difficult table, but looking at the marked skills more frequently achieved by the infants, I could perceive an evolution of the motor function. And I found very interesting the cultural diferences in childcare between the eastern and western parents pointed by the authors, which reflects in infant’s motor development.

I head one concern in the english writing:

Line 271: “We also had only one infant with PIBI after 10-12 months. As the infant had a very mild brain injury, they showed similar motor behavior compared to HFI.” If it is only one infant, I think it should not be written “they”, but he or she.

Author Response

Authors appreciate the reviewer's kind comments. 

We fixed the comment: they  --> the infant (line 272)

Reviewer 2 Report

Thank you for this work

Author Response

Authors appreciate the reviewer's time and comments. 

We corrected one typo at Line 272:

They --> the infant